# Genetic Heterogeneity, Therapeutic Hurdle Confronting Sorafenib and Immune Checkpoint Inhibitors in Hepatocellular Carcinoma

**DOI:** 10.3390/cancers13174343

**Published:** 2021-08-27

**Authors:** Sara M. Atwa, Margarete Odenthal, Hend M. El Tayebi

**Affiliations:** 1Pharmaceutical Biology Department, German University in Cairo, Cairo 11865, Egypt; sara.mahmoud-atwa@guc.edu.eg; 2Molecular Pharmacology Research Group, Department of Pharmacology and Toxicology, Faculty of Pharmacy and Biotechnology, German University in Cairo, Cairo 11835, Egypt; 3Institute for Pathology, University Hospital Cologne, 50924 Cologne, Germany; m.odenthal@uni-koeln.de

**Keywords:** ANGPT-2, cell death, drug resistance, drug transport, eNOS, genetic variants, immune checkpoint, liver cancer, signaling pathways, sorafenib

## Abstract

**Simple Summary:**

Hepatocellular carcinoma (HCC) represents a worldwide health challenge, ranking globally as the third most common cause of cancer-related mortality. Current advancements in the HCC therapeutic armamentarium succeeded in challenging HCC conventional therapy. Systemic therapies including tyrosine kinase inhibitors and immune checkpoint inhibitors (ICIs) come at the forefront of novel HCC therapeutic modalities. However, emerging drug resistance remains an obstacle during HCC therapy. According to the ongoing genomic analysis of HCC, a complex mutational landscape lies behind HCC pathogenesis and hence, affects the response of the tumor to the applied therapy. This review aims at categorizing and summarizing the different resistance mechanisms confronting tyrosine kinase inhibitors, represented by sorafenib, as well as ICIs, during HCC therapy. In addition, giving an insight into how genomic heterogeneity can influence the response of HCC to the aforementioned therapies.

**Abstract:**

Despite the latest advances in hepatocellular carcinoma (HCC) screening and treatment modalities, HCC is still representing a global burden. Most HCC patients present at later stages to an extent that conventional curative options are ineffective. Hence, systemic therapy represented by the tyrosine kinase inhibitor, sorafenib, in the first-line setting is the main treatment modality for advanced-stage HCC. However, in the two groundbreaking phase III clinical trials, the SHARP and Asia-Pacific trials, sorafenib has demonstrated a modest prolongation of overall survival in almost 30% of HCC patients. As HCC develops in an immune-rich milieu, particular attention has been placed on immune checkpoint inhibitors (ICIs) as a novel therapeutic modality for HCC. Yet, HCC therapy is hampered by the resistance to chemotherapeutic drugs and the subsequent tumor recurrence. HCC is characterized by substantial genomic heterogeneity that has an impact on cellular response to the applied therapy. And hence, this review aims at giving an insight into the therapeutic impact and the different mechanisms of resistance to sorafenib and ICIs as well as, discussing the genomic heterogeneity associated with such mechanisms.

## 1. Introduction

Liver cancer ranks globally as the third leading cause of cancer-related mortality [1] with hepatocellular carcinoma (HCC) accounting for 70–90% of total primary liver cancers [2]. HCC is a multi-factorial, multistep and complex process. It is characterized by its rapid infiltrating growth, metastasis in early-stage, high-grade malignancy, and poor therapeutic efficacy [3]. The majority of HCC patients present at an advanced stage (Barcelona Clinic Liver Cancer stage B or higher), albeit implemented surveillance programs, rendering the curative modalities as radical resection, transplantation, and percutaneous ablation procedures ineffective [3]. Accordingly, systemic treatment of advanced HCC has received the spotlight. Molecular targeted therapy has witnessed a major breakthrough with the approval of sorafenib [4,5]. However, both the SHARP and Asia-Pacific trials engaged in sorafenib approval and revealed a significant improvement in survival benefit of advanced HCC patients, yet the increment in overall survival (OS) manifested in approximately 30% of enrolled patients, was only 2 to 3 months [4,5]. A decade thereafter, phase II and III clinical trials were inaugurated, to test several molecularly targeted mediators, though most of such trials showed non-superiority in survival benefits for advanced HCC patients or treatments were accompanied with severe adverse effects [6]. Nevertheless, a number of molecularly targeted mediators exhibited improved clinical efficacy in phase III clinical trials including lenvatinib that was approved as a first-line therapy [7]. In addition, regorafenib, cabozantinib, and ramucirumab were recommended for the second-line setting [8].

Meanwhile, substantial progress in finding alternatives for HCC patients, who were not successfully treated by the first-line setting, was enforced. Recently, treatment with immune checkpoint inhibitors (ICIs) has emerged as a promising therapeutic option for patients with advanced HCC. Physiologically, immune checkpoints are co-inhibitory molecules that act as “brakes” over the immune system to avoid an exaggerated response and restore its activity to a normal level, thus maintaining an immune balance [9]. HCC mostly arises on a background of chronic inflammation-inducing T cell exhaustion, a state that is characterized by an increased expression of co-inhibitory receptors including programmed cell death-1 (PD-1) and cytotoxic T-lymphocyte associated protein-4 (CTLA-4) [10]. Engagement of co-inhibitory receptors with their ligands is one arm of escaping immune surveillance through attenuating the cytotoxic activity of T cells and thus tumor progression [11]. Indeed, this was one rationale for developing immunotherapeutic techniques for HCC [12], as inhibition of immune checkpoints can aid at leveraging the anti-tumor immune response mediated through cytotoxic T cells. Currently, two classes of ICIs, targeting PD-1 or CTLA-4, are clinically approved or still under investigation for safety and efficacy profiles [13].

Despite the gained momentum in advanced HCC therapeutic modalities, the emergence of drug resistance remains an obstacle. An extensive genomic analysis of primary and recurrent HCC has revealed a complex mutational landscape that could be integrated into drug resistance, thus further understanding of the genomic background of HCC is essential to overcome therapeutic challenges [14]. In the present review, we are focusing on the therapeutic impact of sorafenib and ICI treatment strategies, their benefits as well as discussing their mechanisms of resistance together with the related genomic background. To achieve the purpose of the review, research was conducted at the States National Library of Medicine (PubMed). For the search in databases, the descriptors used were: “drug resistance”, “sorafenib”, “immunotherapy resistance” in combination with “HCC”, “Liver cancer” and “genetic variants”, “mutations”. Research papers and published data were reviewed for their relevance to the aim of the review and summarized. Criteria for inclusion were complete, relevant publication, available online, in English, published mostly between 2008 and 2020.

## 2. HCC Conventional Therapy

HCC management is a challenging process owing to the possible complex underlying co-morbidities and tumor extent as well as the severity of liver dysfunction [15]. Accordingly, HCC treatment-decision making is a multiple-disciplinary approach that requires a high level of expertise to achieve an optimum patient outcome [16]. Despite the curative potential of radical treatments such as surgical resection and transplantation, they are recommended solely for early-stage HCC [17], yet more than two-thirds of HCC patients present at an advanced stage [18]. Besides, one major drawback for surgical resection is that almost 70% of patients develop recurrent HCC post-resection [19]. On the other hand, liver transplantation is considered the definitive treatment modality as it not only removes the existing detectable tumor but also removes the unhealthy liver as well as the preneoplastic lesions within the cirrhotic tissues [18]. Percutaneous local ablation procedures as radiofrequency ablation and percutaneous alcohol injection are recommended for small tumors in patients who are ineligible for tumor resection [20]. Moreover, patients with unresectable tumors and no vascular invasion or metastasis, are eligible for transarterial chemoembolization (TACE) [21]. TACE is mostly coupled with targeted delivery of cytotoxic chemotherapeutic agents as doxorubicin, epirubicin, or cisplatin, thus decrease tumor progression and improve OS [22]. TACE is frequently used as a bridging modality to downsize the tumor prior to liver transplantation [23]. Nevertheless, the aforementioned treatment modalities are not effective for advanced HCC, thus targeted treatments have emerged as a promising technique.

## 3. Sorafenib as a Frontline Therapy for Advanced HCC

After the molecular revolution in the 1980s and a better understanding of cancer etiologies, the development of novel therapies targeting specific pathways in cancers has evolved since then. In 1994, a collaboration project was inaugurated between Bayer and Onyx, in which they focused on the discovery of the Ras/Raf/MEK/ERK pathway as a novel therapeutic target. High throughput screening for Raf 1 kinase inhibitory activities has identified a lead compound that was optimized than to give sorafenib [24]. In 2007, the European Medicines Agency and Food and Drug Administration (FDA) approved sorafenib as a first-line treatment for advanced HCC [5]. Since then, several phase 3 clinical trials have studied other drugs compared with sorafenib in the same setting of advanced HCC, yet none of them showed superior outcomes over sorafenib [25,26,27,28].

Sorafenib is an oral multikinase inhibitor that exerts its anti-tumor activity by inhibiting both, tumor cell survival as well as tumor vascularization [29]. Notably, it has been reported that deregulation in Raf/MEK/ERK pathway has a critical role in HCC development [30]. The anti-proliferative activity of sorafenib was manifested by its interruption of the Raf/MEK/ERK pathway by inhibition of Raf serine-threonine kinases [31]. In addition, sorafenib has been demonstrated to mediate anti-angiogenic activity through targeting receptor tyrosine kinases including vascular endothelial growth factor receptor (VEGFR2 & VEGFR3), platelet-derived growth factor receptor (PDGFR) as well as mast/stem cell growth factor receptor (c-Kit) [32].

Two groundbreaking phase III randomized, multicenter, double-blind, placebo-controlled studies were pivotal in the approval of sorafenib therapy for advanced HCC. The Sorafenib Hepatocarcinoma Assessment Randomized Protocol (SHARP) study revealed that sorafenib can prolong the median overall survival (OS) for approximately three months (10.7 vs. 7.9, sorafenib vs. placebo) [5]. In addition to the Asia-Pacific trial that demonstrated a marginal improvement in median OS (6.5 months vs. 4.2 months, sorafenib vs. placebo) [4]. Since then, sorafenib has prevailed as the therapeutic armamentarium against advanced HCC [8]. Nevertheless, it is noteworthy that both studies declared that the OS rate was manifested in approximately 30% of patients [4,5], a modest response that could be attributed to inherent or acquired resistance to sorafenib [33].

### 3.1. Mechanisms of Sorafenib Resistance and the Related Genomic Background

Intratumor heterogeneity is a pivotal reason that leads to the emergence of drug resistance in tumors (Table 1). Cancer drug resistance is classified into two types. The intrinsic drug resistance occurs prior to the drug exposure and allows the resistant cancer cells to proliferate and form a tumor mass insensitive to chemotherapy. While acquired drug resistance occurs after drug exposure, in which cancer cells develop resistant techniques to halt chemotherapy-mediated cytotoxicity [34]. As illustrated in Figure 1, the emerged drug resistance involves a complex of mechanisms that are related to the transport of drugs across the cell membrane, imbalance in the regulation of cell death through apoptosis and autophagy, genetic variations in the molecular targets and pathways as well as other miscellaneous mutations that enhance drug resistance as those occurring in Angiopoietin-2 gene (*ANGPT-2*) and Nitric oxide synthase-3 gene (*NOS-3*).

### 3.2. Sorafenib Resistance and Drug Transport across Cell Membrane

One pillar of drug resistance mechanisms is mediated by integral membrane transporters, implicated in increased drug efflux or reduced drug uptake [45]. Physiologically, human hepatocytes express multiple transporters that are capable of uptaking endogenous substances and drugs across the sinusoidal membrane as well as their efflux into bile [46]. In HCC therapeutic framework, two large families of membrane transporters are involved in drug resistance, the efflux ATP-binding cassette (ABC) transporters and the solute carrier (SLC) superfamily uptake transporters.

#### 3.2.1. ABC Transporters

The ABC transporters are a superfamily of integral membrane proteins that are ubiquitously found in prokaryotes and eukaryotes [47]. A functional ABC transporter typically contains two transmembrane domains (TMD) and two nucleotide-binding domains (NBD) or ATP-binding cassettes [48]. ABC transporters have a mainstay role in determining the bioavailability of a plethora of drugs including anti-neoplastic drugs and thus have a major contribution to drug resistance modulation [49]. In human, ABC transporters family include 51 genes, three of which are pseudogenes, classified into seven families (A–G) based on their gene structure and sequence homology in both TMD and NBD [48,50].

A recent high throughput study by next-generation sequencing has demonstrated the ethno-geographic related genetic variability of ABC transporter family over 138,000 individuals across seven populations. This could be a proposed plausible explanation for the inter-individual differences in drug responses [51]. Of note, clinically related variants of ABC transporter family members, ABCB, ABCC, and ABCG2 were found to be implicated in chemotherapy resistance [52]. Single nucleotide polymorphisms (SNPs) are one type of genetic variability that is extensively studied in ABC transporters. A recent study showed that *ABCB1*, encoding for multidrug resistance protein 1 (MDR1), genetic variant rs2032582 (3435 C > T) is associated with the lowest sorafenib plasma levels in HCC patients and hence, is related to sorafenib resistance [35]. Furthermore, an interesting finding was elaborated by a study conducted on the Chinese population, in which SNPs in *ABCB1* (335T > C, 3073A > C, 3751G > A, and 4125A > C) were associated with the risk of HCC development [53,54,55].

Multidrug resistance-associated protein 2 (MRP2) is the functional transporter protein, encoded by the *ABCC2* gene. The *ABCC2* SNP rs2273697 (1249G  >  A) has been reported to be associated with increased ATPase activity of MPR2, which in turn induces the efflux of sorafenib and thereby sorafenib resistance [36]. In addition, the ABCC2 transporter has been reported to be overexpressed in HCC compared with adjacent healthy livers [56].

According to Huang et al., the breast cancer resistance protein (BCRP), encoded by *ABCG2,* has a significant role in shaping the sensitivity of HCC to sorafenib. Experimental data showed that BCRP mediates the efflux of sorafenib, an action that was hampered through combining sorafenib with a BCRP inhibitor [57]. Tandia et al. have proposed that this developed sorafenib resistance can be attributed to SNP occurring in the *ABCG2* gene. In agreement, both genetic variants of *ABCG2* rs2, 231,137 (34 G > A) and rs2, 622, 604T (1143 C > T) were associated with the low sorafenib plasma levels and improved clinical outcome [35]. It is worth mentioning that genetic polymorphisms of both *ABCB1* and *ABCG2* in relevance to HCC susceptibility, HCC risk of recurrence following liver transplantation [53] as well as a therapeutic response [58] have been thoroughly studied which provide a solid ground for how genetic variability can implicate HCC since its incidence till therapeutic response.

#### 3.2.2. SLC Transporters

The solute carrier (*SLC*) gene superfamily is the second largest family of membrane transporters consisting of more than 400 membrane-bound proteins classified into 65 subfamilies based on sequence similarity [59,60,61]. SLC transporters’ activation relies on the generation of an electrochemical potential difference or an ion gradient to enable the transportation of their substrates across biological membranes [62]. SLC transporters are involved in a plethora of physiological processes including cellular uptake of nutrients, in addition, their role can be broadened to include the uptake of other xenobiotics, including antineoplastic drugs [63]. A seminal study carried out by Schaller et al. using a bioinformatics analysis on next-generation sequencing (NGS) data from approximately 140,000 individuals from seven major human populations has interestingly reported around 204,000 single-nucleotide variants. Of note, most of the SLC variants that were reported were associated with altered drug responses and toxicity phenotypes [64].

Several studies provided consolidated findings of involvement of *SLC22A1* and its protein, organic cation/anion transporter1 (OCT1), in tyrosine-kinase inhibitors’ uptake including sorafenib [37,65]. Herraez et al. have reported the downregulation of OCT1 in HCC followed by reduced sorafenib uptake and poorer drug response. This study has demonstrated that novel variants including R61S fs*10 and C88A fs*16 induce frameshift thus the production of truncated protein and hence, abolished sorafenib sensitivity [37]. Furthermore, Alonso-Pena et al. have identified several *SLC22A1* inactivating variants at a high frequency in HCC such as rs1001179 (c.262T > C), rs34104736 (c.566C > T), rs36103319 (c.659G > T) and rs4646278 (c.859C > G). Moreover, this study has reported other OCT1 mutations including c.262delT (p.Cys88Alafs*16) (*: position of new termination site represented by position number following) and c.181delCGinsT (p.Arg61Serfs*10). Such SNPs and mutations were manifested in lower sorafenib uptake thus, poorer clinical outcomes [38].

Based on a recent study, SLC46A3 varying expression has an impact on sorafenib resistance [66]. According to their findings, SLC46A3 was downregulated in 80% of studied HCC tissue samples compared to non-tumor adjacent tissues. Furthermore, it was reported that tumors expressing lower levels of SLC46A3 had more aggressive phenotypes and a short survival time post-surgery. In addition, ectopic expression of SLC46A3 was accompanied by enhanced sorafenib uptake and hence, ameliorating sorafenib resistance [66].

A recent genome-wide association study performed by Lee et al. in HCC patients receiving sorafenib has reported a relationship between genetic variants in *SLC15A2*, encoding for peptide transporter 2 (PEPT2), and sorafenib responsiveness. Lee et al. proposed that patients with genetic variant rs2257212 in *SLC15A1* with 1048T/T or C/T genotypes displayed a significantly longer progression-free survival than did patients with C/C genotypes [39].

### 3.3. Sorafenib Resistance and Imbalance in the Regulation of Cell Death

Regulated cell death is described as the death response of cells to changes in their microenvironment when other adaptive responses cannot regain cell homeostasis. It can be classified, based on the molecular mechanisms into autophagy, apoptosis, ferroptosis, proptosis, and others [67,68]. However, alterations in the pattern of autophagy and apoptosis have been reported to be involved in sorafenib resistance in HCC.

#### 3.3.1. Autophagy

Autophagy is an evolutionarily conserved process across eukaryotes, which involves an intracellular catabolic degradation process targeting damaged and superfluous cellular components [69]. Physiologically, autophagy plays a critical role in maintaining cellular homeostasis; however, it could have a paradoxical role in cancer based on the tumor cell context [70]. A basic level of autophagy is sufficient for maintaining genomic stability, thus acting as a cancer suppressor, however, once cancer starts; autophagy is highly activated promoting cancer survival under stress conditions [71]. Moreover, autophagy is thought to contribute to the tumor adaptive response under therapeutic stress, which orchestrates the resistance to treatment [72]. Interestingly, some studies reported that sorafenib induces an autophagic-protective response in HCC cells [73,74,75]. Shimizu et al. demonstrated that sorafenib treatment was accompanied by the accumulation of autophagosomes, as evidenced by the conversion of LC3-I to LC3-II protein in in-vitro and xenograft HCC models [74]. Nonetheless, the exact mechanism, underlying autophagy impact on sorafenib sensitivity needs to be further elucidated [76].

Recent cumulative evidence has indicated that genetic variants of autophagy-related genes (ATGs), that are required for autophagosome formation, strongly correlate with HCC development and progression. Furthermore, a recent study, conducted on Chinese patients suffering from HCC, identified five genetic ATG variants (*ATG5* rs17067724, *ATG10* rs1864183, *ATG10* rs10514231, *ATG12* rs26537, and *ATG16L1* rs4663402), that were associated with HCC development. In particular, *ATG10* rs10514231 showed a highly significant association with the risk of HCC development [77]. Noteworthy, in advanced lung adenocarcinoma, two genetic variants in the *ATG10* gene, rs10036653, and rs1864182, were reported to be associated with primary or acquired resistance to a tyrosine kinase inhibitor, gefitinib [78]. Moreover, ATG*5* genetic variants rs510432 and rs548234 are linked to the HCC progression based on chronic HBV infection [79,80].

#### 3.3.2. Apoptosis

Physiologically, the primary cellular response to non-lethal stress is autophagy [81]. However, an exacerbated stress condition that exceeds a critical duration or an intensity threshold can activate an apoptotic program [82]. Apoptosis is a highly regulated form of programmed cell death, playing a major role in maintaining liver volume and cell number during liver development and regeneration [83]. However, impairment of the fine balance between anti-apoptotic and pro-apoptotic proteins has been linked to hepatocarcinogenesis as well as sorafenib resistance [84]. Shimizu et al. reported that the tumor suppressor miRNA let-7 negatively regulates the expression of the anti-apoptotic Bcl-xl protein in HCC. Hence, the ectopic expression of let-7 miRNA leads to repression of Bcl-xl, which in turn results in sorafenib-mediated toxicity [85]. Additionally, it has been reported that co-administration of the BCL-xl molecular inhibitor, ABT-737, with sorafenib showed enhanced anti-tumoral activity in HCC compared to administration of sorafenib alone [86].

p53 is a tumor suppressor protein, that is involved in cell cycle control, apoptosis, DNA repair, and senescence in response to cellular stress [87]. *Tp53* mutations are amongst the most prevalent mutations in HCC that vary based on the tumor etiology [88]. In this sense, aflatoxin B1-induced hepatocarcinogenesis is associated with R249S mutation in the *TP53* gene [89]. It has been postulated that HCC cells that are devoid of functional p53 protein are resistant to sorafenib-targeted therapy [90,91]. A recent study investigated the link between p53 status and the effectiveness of four tyrosine kinase inhibitors; sorafenib, regorafenib, lenvatinib, and cabozantinib in HCC cells. This study included a variety of liver cancer cells such as HepG2 cells with wild-type *TP53*, Hep3B with nonsense-*TP53* mutation, SNU423 with inframe *TP53* gene deletion, Huh7, and SNU449 with *TP53* point mutation. It was postulated then that regorafenib and sorafenib showed high effectiveness in HCC cell lines carrying the wild-type *TP53* gene compared to a decreased anti-proliferative and proapoptotic properties in HCC cell lines that lack or have a mutated *TP53* variant [88].

### 3.4. Sorafenib Resistance Based on Genetic Alterations of Molecular Targets and Signaling Pathways

Being anti-angiogenic and anti-proliferative, sorafenib has a myriad of molecular targets, that genetic polymorphisms and mutations in their expressing genes would affect the response of HCC to sorafenib treatment. The elevated tissue expression and serum levels of VEGFR, a major regulator of tumor vascularization, have been reported to be associated with poor prognosis of HCC patients [92,93]. Moreover, several studies have demonstrated an association between the clinical outcome in HCC patients receiving sorafenib and SNPs in genes encoding for VEGF signaling pathway molecular components [94]. VEGFR2, also known as kinase insert domain receptor (KDR), is a principal member of the VEGFR family that enhances the pro-angiogenic activity of vascular endothelial growth factor subtype A (VEGF-A) [41]. The aberrant function of KDR has been reported to be associated with vascular endothelial cell damage, impaired endothelial cell survival, and abnormal vascular repair [41]. Wang et al. reported a genetic missense variant c.1416A > T (p.Gln472His, rs1870377) in the *KDR* gene which induces impairment in the binding efficiency of KDR to the VEGF-A ligand [95]. In addition, a study on a Chinese HCC cohort demonstrated the association of wild-type allele (AA) of rs1870377 with diminished progression an improved response to sorafenib in comparison to the heterozygous (TA) or homozygous (TT) genotypes [41]. Moreover, Wang et al. reported a reduced binding efficiency of the transcription factor E2F to the *KDR* gene promoter associated with the promoter genetic variant of the *KDR* gene, rs2071559 C > T, due to a subsequent alteration in the E2F binding site [95]. Interestingly, Zheng et al. has deduced that the homozygous genotype for the C allele is associated with a shorter OS in HCC patients treated with sorafenib [41].

In this framework, the ALICE-1 study has been conducted to investigate the impact of genetic polymorphisms in genes encoding for VEGF and the clinical response of HCC patients receiving sorafenib. Based on the findings of this study, it is reported that the combination of VEGF-A allele C of rs2010963 and VEGF-C allele T of rs4604006 has been associated with worsened prognosis of HCC patients receiving sorafenib [42].

Recently, a comprehensive analysis of biomarkers, BIOSTORM, was conducted retrospectively in the setting of a randomized phase 3 STORM study on 83 HCC patients receiving sorafenib compared to 105 receiving placebo. This study has been able to generate a 146-gene panel, composed of 87 "poor prognosis" genes and 59 "good prognosis" genes, which can identify HCC patients who are predicted to benefit from sorafenib. 30% of the enrolled patients have been identified to benefit from sorafenib in terms of recurrence prevention [96]. Moreover, Harding et al. [97] conducted a study using a hybridization capture-based NGS assay designed to target 341 cancer-associated genes in 127 HCC patients, of which 81 received sorafenib. It was reported that mutations predicted to activate the PI3K-mTOR pathway were associated with poor clinical outcomes in sorafenib-treated patients compared to patients without such mutations. However, based on Harding et al. findings, mutations predicted to activate the WNT or MAPK pathway, TP53 pathway, cell-cycle control, and chromatin remodeling showed no impact on clinical outcomes. Furthermore, this study has demonstrated the null effect of VEGFA amplification on clinical outcome improvement, despite being previously addressed as one of the biomarkers for extreme sorafenib responders [97].

B-RAF, a member of the RAS/RAF/MEK/ERK pathway, is reported to play a critical role in the development of hepatocarcinogenesis. In addition, B-RAF is one of the major kinases targeted by sorafenib; however, mutations in B-RAF have been identified as a driver of sorafenib resistance in the HCC context [98]. A case report for a patient with non-small cell lung cancer (NSCLC), harboring BRAF G469R mutation, showed a strong and rapid response to sorafenib [99]. However, another case report for an NSCLC and HCC patient receiving sorafenib has demonstrated the efficacy of sorafenib in managing the lung lesions with BRAF G469V mutation, while no response was observed in the hepatic lesions with wild-type (wt) B-RAF. Hence, this study suggested that an improved response to sorafenib can be manifested in mutated B-RAF tumor cells which are characterized by constitutive activation of the RAF pathway [100].

### 3.5. Sorafenib Resistance and Polymorphisms of eNOS and ANGPT-2 Genes

It is conceivable that HCC is a hypervascular tumor, in which angiogenesis is a complex and multifactorial process, whose main player is VEGF and its downstream signaling pathway [101]. However, other pathways are incorporated in angiogenesis including the angiopoietin (ANGPT)-Tie system. The human ANGPT-Tie system is composed of two membrane-bound type-I tyrosine kinase receptors (Tie-1 and Tie-2) and three secreted ligands (ANGPT1, ANGPT2, and ANGPT4) [102]. Engagement of ANGPT-2 with its receptor TIE-2 induces the receptor phosphorylation and hence, the activation of downstream effectors including SH2 domain-containing phosphatase (SHP2) and p85 subunit of PI3K. SHP2 and PI3K induce the activity of endothelial nitric oxide synthase (eNOS) which mediate nitric oxide (NO) production [102]. Physiologically, NO mediates several angiogenic-related processes and has been reported to play a pro-angiogenic role in tumor vascularization [103].

Recent reports documented that gene polymorphisms of both *eNOS* and *ANGPT-2* genes could be correlated with the clinical outcomes in HCC patients receiving sorafenib [43,44]. In the Italian multicenter, retrospective ePHAS (*eNOS* polymorphisms in HCC and sorafenib) study, the prognostic value of three *eNOS* polymorphisms, *eNOS*-786 T > C in the promoter region, a 27bp variable number of tandem repeats in intron 4 (*eNOS* VNTR 4a/b) and *eNOS* + 894 G > T in exon 7 were analyzed in terms of progression-free survival (PFS) and OS. This retrospective study included two independent cohorts of patients, a training cohort of 41 HCC patients and a validation cohort of 87 HCC patients, all receiving sorafenib. In univariate analysis, training cohort patients homozygous for *eNOS* haplotype (HT1:T-4b at *eNOS*-786/*eNOS* VNTR) had a lower median PFS (2.6 vs. 5.8 months) and OS (3.2 vs.14.6 months) than those with other haplotypes. Moreover, based on multivariate analysis in the validation set, patients homozygous for HT1 had a lower median PFS (2.0 vs. 6.7 months) and OS (6.4 vs. 18.0 months) than those with other haplotypes. In this context, an ePHAS study has reported that the presence of a specific haplotype of *eNOS*-786 (rs2070744) and *eNOS* VNTR polymorphisms may identify a subset of HCC patients who are more resistant to sorafenib [43].

In the literature, a few studies have identified possible predictive markers for the sorafenib therapeutic setting to treat HCC patients. Hence, Llovet et al. reported the predictive value of both VEGF-A and ANGPT-2 in HCC patients, receiving sorafenib. They found that a low plasma baseline concentration of both VEGF-A and ANGPT-2 predicted prolonged survival in HCC patients, whereas elevated plasma baseline concentration of ANGPT-2 was correlated with more aggressive tumors [104]. Furthermore, Miyahara et al. reported that high baseline ANGPT-2 serum level is complemented by the poor clinical outcome and tumor aggressiveness in HCC patients receiving sorafenib [105]. Marisi et al. reported through a multicentric retrospective study which included 135 HCC patients, all receiving sorafenib, that *ANGPT2* rs55633437 TT/GT genotypes were accompanied by a lower median OS and PFS than did patients with other genotypes. Moreover, this study identified an *ANGPT-2* haplotype including rs3739392, rs3739391, and rs3739390, that was associated with lower median PFS and OS in HCC patients receiving sorafenib [44].

## 4. Immune Checkpoint Inhibitors; an Update of HCC Therapeutic Armamentarium

Recently, immune checkpoint therapy has enlightened the shadow of advanced HCC therapeutic armamentarium. The activity of T cell-mediated immunity is defined through a balance between stimulatory and inhibitory signals, which shape the adaptive responses against foreign antigens while avoiding autoimmunity [106]. Physiologically, immune checkpoints function as a negative feedback regulator of inflammatory responses following T cell activation [107]. However, in the HCC context, chronic inflammation is a major player that accompanies the immune response, inducing an exhaustion state of T cells [108]. An elevated expression of inhibitory signals, including CTLA-4 and PD-1, is witnessed in exhausted T cells, limiting their effectiveness [109]. CTLA-4 expression is induced in activated T cells upon stimulatory signals from CD28/B7 and TCR/MHC binding, regardless; it is expressed constitutively in Tregs, playing a critical role in Tregs suppressive functions [110]. Being homologous to CD28, CTLA-4 competes with CD28 for their shared ligands B7-1/2, expressed on antigen-presenting cells (APCs) [111]. Nonetheless, CTLA-4 has a higher affinity for B7 ligands, counteracting the stimulatory signals induced by CD28/B7 and TCR/MHC binding [112,113]. Thus, T cells activation or energy state is defined through an intricate balance between CD28/B7 and CTLA-4/B7 signaling [110].

Furthermore, PD-1, an immunosuppressive receptor, is expressed on dendritic cells (DCs), natural killer cells (NKs), activated T cells, B cells, and monocytes [10]. The main ligand of PD-1, PD-L1, is expressed by multiple somatic cells upon exposure to pro-inflammatory cytokines. Another ligand for PD-1 is PD-L2, which is chiefly expressed on APCs including macrophages and DCs [114]. The binding of PD-1 to its ligands impair a myriad of functional processes for T cells including T-cell proliferation and the production of IL2, interferon-gamma (IFN-γ), and tumor necrosis factor-alpha (TNF-α), thus reducing T cell survival and impairing its immune response. Furthermore, engagement of PD-1 with its ligands would interfere with downstream signaling induced by TCR, and hence reducing T cell activity [115]. Of note, regulation of T cells activity through CTLA-4 is mainly manifested in lymphoid tissues, while T cells are naïve [116]. Nevertheless, PD-1 regulates T cell activity at the effector stage of the immune response, especially in peripheral tissues [115].

HCC is recognized as an immunogenic tumor, whose microenvironment is brimmed with stromal and immune cells with an elevated expression of immune checkpoints, inducing an immunosuppressive microenvironment [117]. In addition, this immunosuppressive tumor microenvironment (TME) is stratified with a tolerogenic liver environment as well as an underlying inflammation [118]. Provoked by these findings, the development of ICIs is an essential requisite for the systemic management of HCC. ICIs represent a profound shift in cancer therapy, as they do not target the tumor cells, but the soldiers of the immune system, T cells. Additionally, the rationale behind ICIs is not to activate the immune system, instead, they eradicate inhibitory pathways that manage tumor cells, escape from immunosurveillance [119].

In 2017, U.S. Food and Drug Administration (FDA) granted accelerated approval to the PD-1 checkpoint inhibitor, nivolumab, followed by the approval of another PD-1 inhibitor in 2018, pembrolizumab for advanced HCC patients who had disease progression or suffered from severe adverse events with sorafenib. Nivolumab approval was granted based on the results of the particular study, CheckMate-040 [120]. This clinical trial was a phase I/II study that included two arms, a dose-escalation arm, and a dose-expansion arm as shown in Table 2. The dose-escalation phase was designed to identify the safety profile of the drug at different dose levels in three cohorts (uninfected subjects, hepatitis C virus-infected subjects, and hepatitis B virus-infected subjects). Whereas the primary endpoint for the dose-expansion phase was objective response rate (ORR). This trial was followed by a phase III study, Checkmate-459 [121], conducted by Bristol Myers Squibb Company, intended to investigate the clinical efficacy of nivolumab compared to sorafenib as first-line therapy for HCC. Although, the primary endpoint for this study, OS, was not statistically significant (median OS of 16.4 months in nivolumab versus 14.7 months in the sorafenib group), yet, nivolumab revealed a tendency towards clinical improvement in ORR and complete response rate as first-line therapy for advanced HCC.

Pembrolizumab received its approval by the FDA based on the findings of the non-randomized, multicenter, open-label phase II study Keynote-224 clinical trial [123]. This clinical trial was conducted on HCC patients intolerant or were progressing on to sorafenib. Keynote-224 reported an overall objective response rate of 17%, stable disease in 44% of the cohort, while 33% showed disease progression. Furthermore, pembrolizumab demonstrated an acceptable safety profile with few adverse reactions. Based on the effectiveness and tolerability of pembrolizumab in advanced HCC, the FDA has approved the priority review application for pembrolizumab in the second-line setting for advanced HCC through a randomized, placebo-controlled phase III study, Keynote 240 clinical trial [124]. This clinical trial investigated the clinical efficacy of pembrolizumab compared to the best supportive care in patients with previously treated advanced HCC. Nonetheless, pembrolizumab failed to achieve the pre-specified statistical criteria of OS and PFS, yet, pembrolizumab reduced the risk of death by 22% and improved PFS compared with placebo. Moreover, a multitude of clinical trials has been conducted on other immune checkpoint inhibitors as illustrated in Table 2.

### Immune Checkpoint Resistance in HCC and Genomic Background

Owing to the recent approval of ICIs as part of the therapeutic paradigm for advanced HCC, data are scarce regarding the genomic alterations associated with refractoriness to ICIs. However, Harding et al. performed a clinical sequence of 127 HCC patients receiving molecular targeted therapy, 31 of which were receiving ICIs [97]. This study provided a solid ground confirming that HCC cold tumors, defined by WNT/CTNNB1 mutations [130], promote immune escape and consequently, resistance to ICIs. Furthermore, two studies [131,132] employed RNA sequence data to classify HCC by gene expression signature involved in immunity. Both studies identified an HCC cluster with β-catenin mutations that were characterized by an immunosuppressive phenotype. On the other hand, Spahn et al. performed a retrospective multicenter analysis for in-depth characterization of responding and non-responding HCC patients receiving PD-1 inhibitors [133]. This study performed NGS in a subset of 15 HCC patients and revealed that 4 patients have alterations in WNT/β-catenin, 1 of which showed shorter PFS than median PFS for the whole cohort. Based on this finding, Spahn et al. deduced that HCC patients could still benefit from ICIs despite alterations in the WNT/β-catenin pathway. Furthermore, it is widely accepted that tumor mutational burden (TMB) is an indicator for the outcome of the ICI therapy [134,135]. It is postulated that tumors with high TMB, have more neoantigens, thus enhanced immune filtration and hence a higher potential to benefit from ICIs [136,137]. However, Xie et al. reported that HCC patients with high TMB showed lower CD8^+^ T cells enrichment than those patients with low TMB and hence poor prognosis [138]. Based on these findings, genomic mutations are reported to have an orchestrating role in HCC immune microenvironment that can define the response to ICIs therapy.

## 5. Conclusions and Perspectives

Despite the breakthrough in the therapeutic paradigm of HCC, drug resistance still represents a burden confronting HCC management. Of note, drug resistance is a complex and dynamic process, consequently, defining the molecular mechanisms underlying drug resistance is an essential approach. Besides, genomic heterogeneity of HCC plays a major role in the intricate response of the tumor to the applied therapy. Hence, this review aims at discussing the different resistance mechanisms confronting sorafenib as well as ICIs. Accordingly, defining predictive biomarkers with high selectivity and sensitivity to guide the rationale use of sorafenib and ICIs for HCC patients. Moreover, defining the mechanisms orchestrating the pharmacological refractoriness of HCC to the applied therapeutic drugs would aid in identifying novel strategies that can conquer cancer cells and hence, improve the outcomes of HCC patients. Furthermore, it is noteworthy that genomic studies of HCC patients can aid in tailoring personalized treatment paradigm for HCC patients through a rational selection of therapeutic drugs. Advancements in the genomic and clinical fields of HCC in the last decade provided hope for substantial improvement in the clinical outcomes of HCC patients.

## Figures and Tables

**Figure 1 cancers-13-04343-f001:**
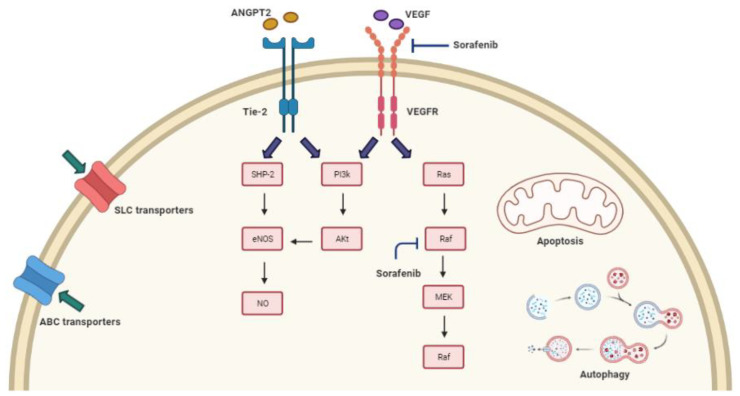
Resistance mechanisms to sorafenib in HCC patients; sorafenib resistance developing in HCC patients can be attributed to a complex of mechanisms, including Transport of the drug across the cell membrane, deregulated cell death mechanisms, the genetic variability of molecular targets and pathways as well as miscellaneous mutations including *ANGPT-2* and *NOS-3* genes. ABC: ATP-binding cassette; SLC: solute carrier; ANGPT-2: angiopoietin-2; Tie-2: type-I tyrosine kinase receptors; VEGF: vascular endothelial growth factor; VEGFR: vascular endothelial growth factor receptor; eNOS: endothelial nitric oxide synthase; NO: nitric oxide.

**Table 1 cancers-13-04343-t001:** Genetic polymorphisms in HCC and their impact on sorafenib efficacy.

Gene	Protein		Genotype	Reference SNP	Consequences	Reference
*ABCB1*	Export Pumps	MDR1	3435 C > T	rs2032582	Reduced sorafenib plasma levels	[35]
*ABCC2*		MRP2	1249G > A	rs2273697	Reduced sensitivity	[36]
*BCRP*		ABCG2	34 G > A	rs2,231,137	Reduced sorafenib plasma levels	[35]
			1143 C > T	rs2,622,604T	Reduced sorafenib plasma levels	
*SLC22A1*	Uptake carriers	OCT1	R61S fs *10	novel	Reduced sensitivity	[37]
			C88A fs *16	novel	Reduced sensitivity	
			c.262T > C	rs1001179	Reduced sensitivity	[38]
			c.566C > T	rs34104736	Reduced sensitivity	
			c.659G > T	rs36103319	Reduced sensitivity	
			c.859C > G	rs4646278	Reduced sensitivity	
*SLC15A2*		PEPT2	1048 T/T & C/T	rs2257212	Prolonged PFS	[39,40]
*KDR*	Drug target	VEGFR2	AA genotype	rs1870377	Improved response to sorafenib and longer TTP	[41]
			CC genotype	rs2071559	Shorter OS	
*VEGF*		VEGF-A	C allele	rs2010963	Reduced OS and PFS	[42]
		VEGF-C	T allele	rs4604006	Reduced OS and PFS	
*NOS3*	eNOS		eNOS−786 TT	rs2070744	Reduced OS and PFS	[43]
			*eNOS* + 894 GG	rs1799983	Reduced OS and PFS	[44]
*ANGPT2*	ANGPT2		TT/GT	rs55633437	Reduced OS and PFS	[44]
			Haplotype (HT2)		Reduced OS and PFS	
			T T G	rs3739392 rs3739391 rs3739390		

OS: overall survival, PFS: Progression-free survival, TTP: time to progression. SNP: Single nucleotide polymorphism. *: position of new termination site represented by position number folowing.

**Table 2 cancers-13-04343-t002:** Clinical trials for immune checkpoint inhibitors as monotherapy in Hepatocellular carcinoma.

Target.	PD-1							CTLA-4	PD-L1	
Drug	Nivolumab			Pembrolizumab		Tislelizumab	Camrelizumab	Tremelimumab	Durvalumab	Atezolizumab
Versus	single-arm	single-arm	sorafenib	single-arm	single-arm	sorafenib	single-arm	single arm	single arm	single arm
Trial name	CheckMate-040 (Dose-escalation arm)	CheckMate-040 (Dose-expansion arm)	Checkmate-459	KEYNOTE- 224	KEYNOTE-240	/	/	/	/	GO30140
NCT number	NCT01658878	NCT01658878	NCT02576509	NCT02702414	NCT02702401	NCT02407990	NCT02989922	NCT01008358	NCT01693562	NCT02715531
Treatment line	First/second	First/second	First/second	Second	Second	First	Second	First/second	First/second	Second
Study phase	I/II	I/II	III	II	III	IA/B	II	II	I/II	IB
Study design	randomized	randomized	randomized	non-randomized	randomized	non-randomized	randomized	non-randomized	N/A	randomized
Primary end points	Safety and tolerability	ORR	OS	ORR	OS / PFS	Safety	ORR/OS at 6 months	Tumor response	Safety	PFS
ORR	15	20	17.6	18	18.3	12.2	32	17.6	10	17
PFS (months)	4.1	4	N/A	N/A	3	2.1	2.1	N/A	2.7	3.4
TTP (months)	3.4	/	7.4	N/A	N/A	N/A	N/A	6.5	N/A	N/A
OS (months)	28.6	15	12.3	12.9	13.9	13.6 (IA) 9.3 (IB)	13.8	8.2	13.2	N/A
DOR (months)	17	9.9	N/A	N/A	13.8	N/A	N/A	N/A	N/A	N/A
Result	accepted safety/tolerability	positive	OS did not reach statistical significance	positive	OS did not reach statistical significance	accepted safety/tolerability	positive	Need further investigation	accepted safety/tolerability	Not effective as monotherapy
Reference	[120]	[122]	[121]	[123]	[124]	[125]	[126]	[127]	[128]	[129]

ORR = objective response rate; PFS = progression-free survival; TTP = time to progression; OS = overall survival; DOR = duration of response; DCR = disease control rate; N/A: not available.

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
