# Peer review of "Genetic Heterogeneity, Therapeutic Hurdle Confronting Sorafenib and Immune Checkpoint Inhibitors in Hepatocellular Carcinoma"

_cancers, 2021, doi:10.3390/cancers13174343_

Round 1

Reviewer 1 Report

A timely review` on genetic heterogeneity and drugs for hcc like sorafenib and immune check point inhibitors

the drug resistance  mechanism are related to multiple causes like mutational landscape, transport of the drug across the cell membrane, desregulated cell death mechanism, genetic variability of molecular targetrs and mutations on angpt-2 and nos-3 genes,.

the authors explained well the clinical trials for immune check points inhibitors in table 2 as monotherapy for HCC like nivolumab, pembrolizumab, tislelizumab, camrelizumab, remelimumab, durvalumab, and atezolimuman.

`

Author Response

Response to Reviewer 1 Comments

Point 1: A timely review` on genetic heterogeneity and drugs for hcc like sorafenib and immune check point inhibitors

the drug resistance mechanism are related to multiple causes like mutational landscape, transport of the drug across the cell membrane, desregulated cell death mechanism, genetic variability of molecular targets and mutations on angpt-2 and nos-3 genes.

the authors explained well the clinical trials for immune check points inhibitors in table 2 as monotherapy for HCC like nivolumab, pembrolizumab, tislelizumab, camrelizumab, remelimumab, durvalumab, and atezolimuman.

Response 1: We would like to thank the reviewer for this comment and we do appreciate this kind support.

Reviewer 2 Report

The authors have written a very excellent review  and  very useful for all scientists working in the field, basic researchers and clinicians.

I ask the authors to check one information on the cancer death of HCC. In the simple summer the authors report HCC as the fifth most common cause of cancer related mortality. At the beginning of Introduction the authors report HCC as the fourth leading cause of cancer related mortality and cite Bray F et al 2018, Ref 1 . But in this article , HCC seems to be the third cause of  cancer death ( after lung and stomach cancer )  .

Author Response

Response to Reviewer 2 Comments

Point 1: The authors have written a very excellent review and very useful for all scientists working in the field, basic researchers and clinicians.

I ask the authors to check one information on the cancer death of HCC. In the simple summer the authors report HCC as the fifth most common cause of cancer related mortality. At the beginning of Introduction, the authors report HCC as the fourth leading cause of cancer related mortality and cite Bray F et al 2018, Ref 1. But in this article, HCC seems to be the third cause of cancer death (after lung and stomach cancer) 

Response 1: We would like to thank the reviewer for this comment and we do appreciate this kind support.

Regarding the worldwide mortality rank of HCC, it is updated in the simple summary section (line 13) and in the introduction section (line 40) to be ranked the third according to the cited paper.
